# Biological Applications of Synthetic Binders Isolated from a Conceptually New Adhiron Library

**DOI:** 10.3390/biom13101533

**Published:** 2023-10-17

**Authors:** Claudia D’Ercole, Matteo De March, Gianluca Veggiani, Sandra Oloketuyi, Rossella Svigelj, Ario de Marco

**Affiliations:** 1Lab of Environmental and Life Sciences, University of Nova Gorica, Vipavska cesta 13, Rožna Dolina, 5000 Nova Gorica, Slovenia; claudia.dercole@ung.si (C.D.); matteo.demarch@ung.si (M.D.M.); oloketuyisandra@gmail.com (S.O.); 2Department of Pathobiological Sciences, School of Veterinary Medicine, Louisiana State University, Baton Rouge, LA 70803, USA; gianluca.veggiani@gmail.com; 3Department of Agrifood, Environmental and Animal Science, University of Udine, via Cotonificio 108, 33100 Udine, Italy; rossella.svigelj@uniud.it

**Keywords:** affimers, adhirons, C-Reactive Protein, SpyCatcher, phage display, in vitro panning

## Abstract

Background: Adhirons are small (10 kDa) synthetic ligands that might represent an alternative to antibody fragments and to alternative scaffolds such as DARPins or affibodies. Methods: We prepared a conceptionally new adhiron phage display library that allows the presence of cysteines in the hypervariable loops and successfully panned it against antigens possessing different characteristics. Results: We recovered binders specific for membrane epitopes of plant cells by panning the library directly against pea protoplasts and against soluble C-Reactive Protein and SpyCatcher, a small protein domain for which we failed to isolate binders using pre-immune nanobody libraries. The best binders had a binding constant in the low nM range, were produced easily in bacteria (average yields of 15 mg/L of culture) in combination with different tags, were stable, and had minimal aggregation propensity, independent of the presence or absence of cysteine residues in their loops. Discussion: The isolated adhirons were significantly stronger than those isolated previously from other libraries and as good as nanobodies recovered from a naïve library of comparable theoretical diversity. Moreover, they proved to be suitable reagents for ELISA, flow cytometry, the western blot, and also as capture elements in electrochemical biosensors.

## 1. Introduction

In the last thirty years, the research community has looked actively for reagents that could represent valid alternatives to conventional antibodies and their fragments. Camelid heavy-chain only IgGs were identified in 1993 [1], and their unique variable domain (VHH or nanobody) was immediately considered a potential substitute for the scFv format generated starting from conventional IgGs. The recombinant nature of such molecules simplifies their engineering, in silico optimization, cloning, and production as reagents fused to different application-friendly tags, and their small size has been exploited in super-resolution microscopy [2,3] and for nanostructure functionalization [4]. Large nanobody naïve libraries prepared to exploit the natural diversity present in llama herds provided useful binders [5], but the constantly decreasing costs of gene synthesis combined with the wider availability of paratope structural information enabled to generate totally synthetic collections of more sophisticated designs in terms of framework sequence, scaffold structure, and loop variability and geometry [6,7,8,9].

In parallel with the development of nanobody libraries, other single-domain scaffolds have been exploited for preparing collections of molecules with local hypervariable surfaces that could work as an alternative to antibodies and antibody fragments but preserve high antigen selectivity and elevated affinity [10,11]. Despite the large number of proposed scaffolds that have been described and proved to be functional, only a few of them, such as affibodies or DARPins, became widely established reagents [12,13]. The intrinsic quality of such molecules has been indubitably a critical selection factor, but it should not be underestimated that not all the groups that proposed a new scaffold had the possibility to perform the long-lasting, systematic validation and dissemination work necessary to convince the community of the reliability and utility of the ligand. In other cases, the existence of the patents might have discouraged the researchers from investing time and resources in testing potentially interesting molecules. We started looking among proposed scaffolds for those with biophysical characteristics that could represent an advantage with respect to nanobodies, and our attention was captured by adhirons [14] because of their following characteristics: extremely high stability; highly elevated yields; and simple production strategies due to the lack of cysteines in their original sequence. Furthermore, their extremely short sequence should also reduce the computational requirements for in silico optimization. The adhiron scaffold is a synthetic protein designed as a consensus sequence inferred from those of plant cystatins. These proteins are structurally similar to the human stefin A, another protein that has been proposed as a scaffold for peptide presentation [15], later called affimer [16], a registered trademark of Avacta Group. At present, the two scaffolds are collectively referred to as affimers [17], a somehow misleading convention that we tried to overcome while keeping the terms adhiron and affimers separate. According to the literature, adhirons/affimers are still almost exclusively isolated in the labs that originally designed and described them [14,17], but since the data accumulated over the years clearly indicated their functionality [18,19,20], we decided to evaluate the challenge represented by the preparation of a new adhiron library and its panning. In this perspective, we also tried to obtain binders that could be directly compared for their biophysical features with already available nanobodies isolated for the same antigens [21] and then evaluated their performance in a set of different applications. 

## 2. Materials and Methods

### 2.1. Library Preparation

The phage display adhiron library was obtained by combinatorial site-directed mutagenesis of a sequence in which the adhiron coding sequence was flanked by the *Nco*I and *Not*I restriction sites for subcloning into compatible bacterial expression vectors [22,23].

Mutagenesis of the loops connecting β-strands 1 and 2 and β-strands 3 and 4 was performed using the following degenerate primers: 5′-CAAAGCAAAAGAGCAAXXXXXXXXXACGATGTATTATTAAC-3′ together with 5′-CAAGGTCTGGGTAAAAXXXXXXXXXAATTTTAAGGAACT-3′, where X denotes NNK codons with N = A/G/C/T and K = G/T; and 5′-CAAAGCAAAAGAGCAAZZZZZZZZZACGATGTATTACTTAAC-3′ in combination with 5′-CAAGGTCTGGGTAAAAZZZZZZZZZAATTTTAAGGAACTTC-3′, where Z stands for NDT; N = A/G/C/T, D = A/G/T, enabling to encode R,N,D,C,G,H,I,L,F,S,Y,V amino acids (Appendix A). The adhiron library was developed via Kunkel mutagenesis, as previously described, by simultaneously mutating both loops with equimolar combinations of all degenerate primers. The adhiron sequences were finally cloned in a pHEN-derived phagemid that has been designed ad hoc to be in-frame with a secretion and multiple-tag sequences, as detailed in Figure 1a.

### 2.2. Protoplast Preparation

*Pisum sativum* plants were grown in a climatized chamber at 25 °C with a light/dark photoperiod of 16:8 hours. Intra-veining strips were cut out of leaves from 2-week-old plants (1.5 g) and incubated for 210 min at 30 °C with gentle shaking (70 rpm) in 10 mL of 1 M mannitol, 100 mM MES-KOH, pH 5.6, 0.37% *w/v* macerozyme, and 1.5 % *w*/*v* cellulase (Duchefa Biochemie, Haarlem, The Netherlands). The digestion reaction was stopped by adding 2 mL of 2 mM MES, 154 mM NaCl, 125 mM CaCl_2_, 5 mM KCl, pH 5.7, and the cell suspension was filtered through a polypropylene mesh (pore diameter: 105 µm; tissue thickness: 121 µm—polyester Woven filter, Spectrum Labs, Rancho Dominguez, CA, USA). Digested cells were washed twice by resuspending them in 2 mL of 4 mM MES, 0.4 M mannitol, and 15 mM MgCl_2_, pH 5.7. Finally, they were pelleted by gentle centrifugation (3 min at 100× *g*), resuspended in 1 mL of the same buffer, counted, and stored at 4 °C after having confirmed their integrity at the microscope.

### 2.3. Panning and Screening of Adhiron Phage Display Library on Whole Protoplasts

The panning strategy applied in combination with soluble antigens followed the same steps and conditions previously used when nanobody libraries were screened [24]. In contrast, the specific characteristics of the phagemid enabled us to improve the protocol originally developed for panning on whole cells [25] and to apply it to select the adhirons specific for protoplasts. Briefly, the adhiron phage library was first depleted from binders recognizing plastics and coating material and then incubated with the protoplasts (1 million). After extensive washing, the bound phages were eluted in 0.1 M glycine-HCl pH 2.2, 1 mg/mL BSA and amplified before starting a second panning cycle. The finally recovered clones were screened by the flow cytometry against purified protoplasts (50,000) using an anti-ALFAtag-mRuby3 nanobody reactive toward the corresponding tag displayed in the nanobody tag sequence and recording the fluorescent emission signal at 592 nm. The analysis was performed in triplicate, and clones were considered positive when showing a fluorescence increase > 3.

### 2.4. Panning on Soluble Antigens

The adhiron library was panned against C-Reactive Protein (CRP), SpyCatcher (version 002, [26]), and the Receptor Binding Domain (RBD), the domain of the SARS-CoV-2 spike protein. CRP was bought from Merck—Darmstadt (ML-AG723); SpyCatcher and RBD were produced in-house, as previously described [27,28]. An aliquot of 100 µL of the adhiron phage display library was first depleted against 50 µL of M-450 epoxy magnetic beads (Dynabeads) coated with 500 µg of bovine serum albumin (BSA). In the case of the panning against SpyCatcher, a second depletion step was performed using the same amount of beads coated with 1 mg of a fusion construct formed by the irrelevant anti-HER2 control nanobody A10 and the fluorescent protein mClover3. The unbound library fractions were then incubated in the presence of 50 µL of magnet beads coated with 200 µg of antigen. After two rounds of panning and extensive washing in PBS/PBST plus milk, as described before [23], bound phages were eluted in 0.1 M glycine, pH 2.2. In the case of the anti-SpyCatcher and RBD selection, single clones (92) were finally screened by phage ELISA using anti-M13-horseradish peroxidase (HRP)-conjugated antibodies (Thermo Fischer). The signal of the target antigen SpyCatcher-mClover3 was compared with those obtained with A10-mClover3 and BSA, using 2% skimmed milk in PBS for plate coating and PBS+0.1% Tween (PBST) 20 for washing. Unique sequences were aligned using Clustal Omega. The recovery of the anti-CRP positive clones was performed by screening adhirons obtained by direct expression from the phagemid in not suppressor *E. coli* strain BL21. First, the phagemid was extracted from the TG1 cells and was used to transform BL21 cells, and the resulting single colonies were cultured at 37 °C in 1.5 mL of Terrific Broth medium supplemented with 0.1% (*w*/*v*) glucose, 100 μg/mL ampicillin, and 2 mM MgCl_2_ in a deep-well plate. IPTG (0.2 mM) was added at an OD_600_ of 0.6; the temperature was decreased to 25 °C, and bacteria were grown overnight. Periplasmic proteins were released by osmotic shock, as previously described for recombinant nanobodies [29]. The supernatant was then used for an ELISA-based screening in 96-well plates. Wells were coated overnight at 4 °C with 0.25 µg/well of the antigen in 400 µL and then saturated with 2% BSA in PBS. After washing in PBST, 100 µL of periplasmic fraction were added, and samples were incubated for 2 h at room temperature and finally washed again in PBST. Controls were performed in the presence of 2% BSA in PBS. Home-made SpyCatcher-mClover3 (100 µL at 1 mg/mL) was added, and after 1 h incubation at room temperature, the wells were washed in PBST and then filled in with 400 µL of PBS. The absorbance at 535 nm was measured in a Tecan microplate reader (Infinite F200—Männedorf, Switzerland), and clones were considered positive when their signal over the background was higher than 5.

### 2.5. Adhiron Expression, Purification, and Characterization

Alternative adhiron sequences designed for preliminary evaluation were ordered as synthetic genes from Twist Bioscience (San Francisco, CA, USA). Both such constructs and adhiron sequences isolated by panning were cloned into modified pET-14a (+) vectors to obtain fusion constructs consisting of a C-terminal 6xHis tag, and an additional tag among SpyTag, free cysteine, AviTag, fluorescent proteins mRuby3 and mClover3, rabbit Fc domain, murine µ domain, and monomeric ascorbate peroxidase—APEX2) [23,30]. The isoelectric point (pI) of the different constructs was calculated by applying the ProtParam software [31]. Expression vectors were transformed in both wild-type *E. coli* BL21(DE3) cells and in the same cell strain co-transformed with a plasmid containing the sequence of a sulfhydryl oxidase (SOX). In such a way, it was possible to compare the production under conditions that either impaired or promoted the formation of disulfide bonds. In the case of SOX co-expression, the bacteria were grown in LB medium at 37 °C and 210 rpm, and the accumulation of SOX, necessary to obtain the formation of disulfide bonds in the cytoplasm [32], was induced by adding 0.2 % (*w*/*v*) of arabinose when the bacterial culture reached the OD_600_ of 0.4. Simultaneously, the temperature was decreased to 20 °C, and 0.1 mM IPTG was added when the OD_600_ reached 0.8. The constructs grown in wild-type bacteria were directly induced with IPTG at the same conditions of temperature and concentration indicated above. Bacteria were grown overnight at 20 °C, and then, cells were pelleted by centrifugation (30 min at 4500× *g*) before being lysed in four volumes of 50 mM Tris-HCl pH 8.0, 500 mM NaCl, 5 mM MgCl_2_, and 15 mM imidazole by alternating three cycles of freezing/thawing. After the addition of DNAseI (3U) and lysozyme (100 μg/mL), lysates were incubated for 30 min at 4 °C, sonicated (amplitude 80%, 1 min pulse, 1 min off), and finally centrifuged at 4 °C (30 min at 13,000× *g*). The supernatant was injected at a flow rate of 1 ml/min into a Talon affinity Hi-Trap column (Cytiva—Marlborough, MA, USA), previously equilibrated with buffer A (50 mM Tris-HCl pH 8.0, 500 mM NaCl, 15 mM imidazole), washed with the same buffer, and finally eluted in buffer B (buffer A + 500 mM imidazole) using an FPLC system (Cytiva). Eluted samples were desalted by means of a Hi-Trap column (Cytiva) in appropriate buffers chosen according to protein pI, solubility index (variable salt concentration), and the presence of free cystines (TCEP addition). The resulting samples were evaluated by SDS-PAGE and analytical gel filtration (Superdex 75 Increase 5/150 GL, Cytiva). When necessary, a preparative gel filtration step (Superdex 200 10/300 GL, Cytiva) was used to remove minor contaminants and aggregates. As an alternative purification strategy, the supernatant fractions were incubated for 15 min at temperatures between 50 °C and 100 °C and then for 20 min on ice to induce the selective precipitation of contaminant bacterial proteins (thermal purification, [33]). Sample monodispersity was evaluated by analytical gel filtration and UV scan between 250 and 330 nm. Protein concentration was assessed by measuring the absorbance at 280 nm, and the presence of contaminants was evaluated after sample separation by SDS-PAGE. 

Western blot analysis was performed as follows: CRP (AG723 Sigma-Aldrich) aliquots of 4.5 μg were separated by SDS-PAGE and electrophoretically transferred onto nitrocellulose membrane (VWR, Cat. No. 10600001) for 1.5 hours at 0.3 A using a Hoefer SemiPhor Semi-Dry Transfer Cell. The transfer efficacy was verified by staining the membrane with Ponceau S Staining Solution before the addition of the blocking solution (5% milk in PBS, pH 7.4). After overnight incubation at 4 °C, the membrane was cut into strips corresponding to the loaded CRP aliquots. Each nitrocellulose strip was incubated for 1.5 hours under constant agitation in the presence of 20 μg of anti-CRP adhirons diluted in PBS pH 7.4, 5% milk, 0.05% Tween-20. Membranes were washed in PBS pH 7.4, 0.05% Tween-20 for 30 min before the addition of anti-HisTag HRP secondary antibodies (Novus Biologicals, NB100-63173), diluted 1:1000 in PBS pH 7.4, 5% milk, 0.05% Tween-20. Samples were incubated for 1 hour under constant agitation and then washed for 1 hour as indicated above before the addition of the SuperSignal™ West Femto Maximum Sensitivity Substrate chemiluminescent substrate (Thermo Fisher, Cat. No. 34094). Membranes were imaged with the UVITEC Cambridge chemiluminescence imaging system.

### 2.6. Binding Affinity Determination

Surface plasmon resonance (SPR) was performed, as previously described, using a Biacore T100 (GE Healthcare, Uppsala, Sweden). 

The assessment of the adhiron binding affinity by ELISA was performed as follows. Soluble antigens (CRP and SpyCatcher) were diluted to a final concentration of 10 µM using 50 mM Na_2_CO_3_/NaHCO_3_ pH 9.5, and Nunc MaxiSorp Immunoplates (Thermo Fisher Nunc—Waltham MA, USA) were coated with 100 μL/well of each antigen before being incubated overnight at 4 °C. In the case of protoplasts, plates were first coated with 5 µg of Spy-Tagged adhiron resuspended in 100 µL of 0.2 M sodium carbonate/bicarbonate buffer pH 9.5 and incubated overnight at 4 °C. After 3 washing steps in PBS pH 7.4, 40,000 protoplasts were added to each well. Plates were washed three times in PBST buffer (0.05% Tween 20 in PBS pH 7.4), blocked with 1% BSA, incubated for 1 h at room temperature (RT) in PBST and then 4 h at RT in the presence of serial dilutions of adhiron-APEX2 fusion construct (100 μL/well, concentrations ranging between 4 µM–23 pM) in PBS + 1% BSA. After three washing steps in PBST, 50 μL of freshly prepared substrate solution (Amplex, Thermo Fisher) was added to each well, and the fluorescent signal was measured at 560 nm using a microplate reader (Tecan—Männedorf, Switzerland). Three replicates were performed for each concentration, and after background subtraction, the data were normalized by subtracting the background signal and analyzed with Prism9 (GraphPad Software—Boston, MA, USA) using either Hill slope fitting or the single binding model to recover the EC50 values.

### 2.7. Differential Scanning Fluorimetry (DSF) Assays

Thermally induced protein unfolding assays were used to determine the melting temperatures (Tm) of adhirons. The optimal signal-to-noise ratio between adhiron and dye (0.1 μg/μL of protein sample and 5x of SYPRO orange, Thermo Fisher) was identified in preliminary tests performed using different concentration combinations. Adhirons were diluted in buffer A (50 mM Hepes pH 6.5) or B (50 mM PBS pH 7.4) according to pI and further mixed with the dye (SYPRO orange, Thermo Fisher Scientific); 50 μL/sample was added to MicroAmp 96-Thermowell PCR plates (Thermo Fisher). The effect of additives (100 mM NaCl, 5 mM DTT, or 5 mM EDTA) was also tested. Reactions were carried out in a Light-Cycler 480 II real-time qPCR (Roche—Basel, Switzerland) with a temperature gradient ranging from 25 °C to 95 °C and an amp rate of 0.01 °C/s. Fluorescence was detected using an excitation wavelength of 465 nm, an emission of 580 nm with continuous acquisition mode, and recording 100 acquisitions per 1 °C. Melting temperatures were determined with the Origin 8.1 software (OriginLab Corporation—Northampton, MA, USA) using the Boltzmann function.

### 2.8. Biosensor Preparation

The immunocapture element of an electrochemical impedance biosensor was prepared by conjugating SpyCatcher to the detection surface to covalently bind the E7-SpyTag anti-CRP adhiron construct, and electrochemical measurements were carried out as previously described [24]. Electrochemical Impedance Spectroscopy (EIS) was performed at open circuit potential, and Nyquist plots of EIS data were fitted with the equivalent circuit using ZView software. Cyclic voltammetry (CV) was conducted at a scan rate of 20 to 200 mV s^−1^ to characterize the response of the bare electrode. Measurements were performed at room temperature in a 0.1 M KNO_3_ solution containing 5 mM equimolar K_3_[Fe(CN)_6_]/K_4_[Fe(CN)_6_] at a potential range of −0.7 to +0.3 V and a scan rate of 20 mV s^−1^. As an alternative, Dropsens screen-printed carbon electrodes (Metrohm, Milan, Italy) were used to perform electrochemical measurements with an Autolab PGSTAT204 potentiostat (Metrohm, Milan, Italy) managed by Nova software version 3.2 and connected to the SPCE by means of a CAC connector cable (Metrohm, Milan, Italy). BSA, 3,3′,5,5′- tetramethylbenzidine (TMB) liquid substrate, and biotin were purchased from Sigma-Aldrich (Milan, Italy). Streptavidin-HRP conjugate was purchased from Merck (Milan, Italy). Ultrapure water (R > 18 MΩ) was obtained by means of an Elga Purelab Flex 4 system (Veolia Water Technologies, Milan, Italy) and used for the preparation of buffer solutions. Ten μL of streptavidin (1 mg/mL) were immobilized on the bare carbon electrode by adsorption at 4 °C overnight, and subsequently, the surface was blocked for 30 min with a solution of PBS pH 7.4 containing 0.5% BSA. Biotinylated SpyCatcher (1 μM) resuspended in PBS pH 7.4 was incubated for 30 min at room temperature to favor its immobilization on the coated streptavidin. Residue-free streptavidin was blocked with 0.5 µM biotin solution in PBS pH 7.4 for 30 min. Next, solutions at increasing concentrations (between 25 and 500 nM) of biotinylated anti-SpyCatcher G5 adhiron in PBS pH 7.4 were incubated for 30 min. The binding was quantified by incubating streptavidin-HRP at 0.75 μg/mL for 10 min, and the electrochemical transduction was obtained by adding 60 μL of TMB solution. The reduction current was measured by chronoamperometry at 0 V after 1 min of the enzymatic reaction. The corresponding curves were fitted using the Hill equation.

## 3. Results and Discussion

### 3.1. Design of the Adhiron Phage Display Library

Adhirons have been originally engineered by grafting two sequences of nine amino acids each at the place of the short linkers connecting β-strands in the native phytocystatin consensus protein [14]. Specifically, the first loop took the place of the two terminal residues (VV) of the first β-strand plus the two residues (AG), connecting it with the second strand, whereas the second loop substituted for the three amino acids linking β-strands 3 and 4 (Figure 2, amino acids are evidenced in red). Initially, we wished to test the capacity of adhirons and adhiron variants to accommodate nanobody CDRs in their structure, namely, to verify if CDR grafting could affect their stability and preserve at least part of their original binding capacity. 

The first trial was performed by grafting the CDR1 and CDR3 of an anti-HER2 nanobody in place of the two artificial adhiron loops since the residues present in such sequences compose most of the main paratope of this nanobody. The CDR1 (ATSNIS) is preceded by serine and glycine, both involved in the antigen binding and, therefore, at the place of the first loop, it was possible to insert a sequence that preserved three of the four residues present in the original phytocystatin sequence (VVSGATSNIS, Figure 2). The nine-residue long CDR3 took the exact place of the adhiron second loop (Figure 2). The construct (Adh1) was expressed as a fusion to eGFP in high quantity (11.6 mg/L) and showed a binding constant (*K*D = 36 nM, Figure 2 and Appendix A) comparable to that of the original anti-HER2 nanobody A10 (8.1 nM). To further investigate whether the adhiron scaffold can accommodate different CDRs, we replaced the two canonical adhiron loops with the CDR3 of two different anti-HER2 nanobodies of comparable affinity (A10 and C8, [23]). Both nanobody-derived CDR3s were nine amino acids long, and we also assessed whether the presence of only portions of the original paratopes (lacking CDR1 and CDR2 of both nanobodies) could be compensated by the avidity effect (Figure 2 and Appendix A). A second adhiron (Adh2) was recombinantly expressed as fusion to eGFP with good production yields (6 mg/L), and SPR analysis revealed that CDR3 grafting enabled HER2 binding with high affinity (*K*D = 31 nM). Finally, we assessed a shorter version of Adh2, obtained by removing the unstructured 10-amino-acid long sequence at its N-terminus (Adh3, Figure 2 and Appendix A). While the new construct was produced at high yields (12 mg/L), its binding capacity to HER2 was completely abrogated. Since the aim of these preliminary experiments was to identify a robust scaffold rather than investigate the role of the adhiron N-terminal portion, this construct was no longer considered. Altogether, the collected data were sufficient to consider the standard adhiron framework as a robust scaffold for the creation of a novel phage display library by hypermutating the two nine-residue loops.

The new library construction was performed by combinatorial site-directed mutagenesis of a phagemid designed for the display, on the major phage coat protein pIII, of the adhiron sequence fused to a set of tags at its N-terminus. After a secretion leader sequence, the phagemid presents a 6xHis tag, suitable for affinity purification, an ALFA tag, and a SpyTag (Figure 1a). These tags were included in the design for facilitating affinity purification, as well as rapid and effective detection and/or bioconjugation (anti-ALFA tag nanobody and SpyCatcher, respectively [34,35]) with binding partners. Such constructs can be produced inexpensively and at scale. The *Nco*I and *Not*I restriction sites were inserted for downstream subcloning into expression vectors. The adhiron library was cloned via Kunkel mutagenesis, and the hypermutated loops contained both NKK and NDT codons, a strategy that enabled the presence of all 20 amino acids (Figure 1b). Unlike other conventional adhiron libraries, our randomization strategy allowed the presence of cysteine residues in their loops, as non-canonical cysteines within the CDRs of the heavy chain of human antibodies significantly expand the diversity of the paratope space [36]. We opted for this design that potentially enables the formation of disulfide bonds to increase the overall structural diversity of the library paratopes, providing interaction surfaces with a geometry probably missing in standard adhirons [14]. In our opinion, this advantage could compensate for the drawback represented by the folding requirements of cysteine-containing proteins. Such a strategy enabled us to construct a novel phage display library containing 9 × 10^9^ unique adhiron clones.

### 3.2. Panning and Binder Characterization

The library was validated by panning against antigens with different characteristics, such as CRP, which forms a large pentamer, and the small SpyCatcher domain (10 kDa). CRP is a biomarker that requires accurate monitoring in several pathologies [37], whereas SpyCatcher, besides being a widely used reagent, derives from a portion of the *S. pyogenes* FbaB domain, which is relevant for the invasion of eukaryotic cells [38]. Furthermore, the library was panned against the receptor binding domain (RBD) of the SARS-CoV-2 spike protein and against pea leaf protoplasts. 

To our knowledge, this is the first report of panning on plant protoplasts using a library of recombinant ligands, and this approach might represent a convenient method for the isolation of reagents for plant membrane biomarkers for which conventional antibodies are usually scarcely available. Moreover, similarly to our work with whole mammalian cells [25], we used the novel phage display library for selection on protoplasts and combined it with flow cytometry for rapid screening of binding adhirons. However, protoplasts’ autofluorescence in the whole range of visible light is a major issue for color-based discrimination methods. Therefore, several fluorescent proteins were compared to identify the one enabling the highest signal-to-noise ratio. The combination of the anti-ALFAtag nanobody fused to mRuby3 provided the best specific signal and was, thereby, used in our panning strategy. Protoplasts are fragile structures undergoing rapid deterioration and, consequently, represent a difficult target for effective panning. Nevertheless, by employing this approach, we successfully identified 14 unique clones capable of specific protoplast binding. We focused on four clones (A12, B01, B02, E12) that displayed the highest signal-to-noise ratio in flow cytometry (Appendix A) and noticed that they shared a high homology of the variable loops and a cysteine content superior to that observed in clones isolated from other panning procedures (see below). These four adhirons were subcloned in bacterial expression vectors in order to produce them as fusion proteins with APEX to use in ELISAs. All adhirons were readily expressed, as shown by the example provided in Figure 3a, where E12 could be purified at homogeneity (SDS-PAGE), was monodispersed (gel filtration), and had no apparent contamination (UV scan and 268/280 ratio). The apparent binding strength of each clone was assessed by an optimized sandwich ELISA assay, in which protoplasts were immobilized onto the wells of a 96-well plate via binding to capture adhirons. Plate-immobilized protoplasts were detected by using soluble adhirons fused to mRuby3 (Figure 3b). All four adhirons allowed specific protoplast binding, with A12 and E12 showing the highest fluorescent intensity (Figure 3b), but the high autofluorescence background tended to saturate the signal, impairing precise measurements over a wide range of concentrations. Therefore, A12 and E12 binding affinity was estimated by direct-binding ELISAs using different concentrations of adhirons fused to the APEX enzyme (Figure 3c). The EC50 values of A12 and E12 were 2.32 ± 0.01 and 1.06 ± 0.88 µM, respectively. The results confirmed the adhiron specificity for protoplast surface antigens and suggested that the selected adhirons would benefit from affinity maturation campaigns to improve binding. In this perspective, the experience accumulated with antibody fragments, as well as the highly modular design of our library, will be highly useful to optimize the available adhiron sequences [39].

In parallel with the technically very demanding panning and screening on protoplasts, we tested the library for its capacity to provide binders specific for soluble antigens that have already been used for nanobody panning with different success rates. Recently, we described a pool of nanobodies against CRP with affinities in the nM range [24]. The same antigen has been used as well to isolate affimers with affinities in the µM range [16]. We challenged our adhiron library against CRP, and the resulting clones were screened by ELISA, exploiting the SpyTag fused to the adhiron sequence and reacting it with SpyCatcher-mClover3 fusion protein. The fluorescent signal intensity was measured at 535 nm (Appendix A), and clones were considered positive when their fluorescence was at least five times higher than the background. Thirty-nine positive ELISA hits corresponding to 10 unique sequences were recovered by screening 92 randomly selected clones obtained from as little as two rounds of panning, highlighting the robustness of our library (Appendix A). Among these, four clones had no cysteine, one had two cysteines, and five had a single cysteine. Only one sequence (G6) presented a few mutations in the framework, probably introduced during the amplification steps. Subcloning the isolated adhirons as APEX fusions was successful for seven constructs that were expressed, purified, and characterized similarly to the example reported in Figure 3a. Their EC50 values were calculated by measuring the APEX enzymatic activity at decreasing antigen concentrations (Figure 4a and Table 1). Most of the clones had an apparent affinity in the low nM range, and although the measurement methods were different, the data indicated that the anti-CRP adhirons were at least as good as the nanobodies and significantly better than the affimers previously isolated against the same antigen [16,24]. In a preliminary test, the anti-CRP adhiron E7 (1 µg/mL) proved that it could be successfully exploited as capture elements in an electrochemical impedance biosensor, confirming the functional versatility of isolated adhirons across various experimental methods (Appendix A). 

The library was panned against the SpyCatcher domain as well, taking into account that some not specific binding could happen between the SpyTag present on the phage and the antigen. In contrast to the large CRP multimer, SpyCatcher002 is a 10 kDa domain that offers a limited variety of binding surfaces. No suitable nanobody was identified in a previous panning performed with the same phage-display library successfully used to isolate the anti-CRP ligands. This antigen represented consequently also an interesting model to evaluate whether the alternative geometry of the adhiron paratopes could be more successful in favoring productive interactions with SpyCatcher002. After two rounds of panning, 92 clones were randomly screened by conventional phage ELISA, and 16 showed high selectivity for the target. Sequencing analysis revealed the presence of six unique adhirons, of which two had no cysteine, and four possessed two cysteines in their loops (Appendix A, Appendix A). Adhirons B1 and F9 were retrieved in multiple copies (10 and 2, respectively). The binding between SpyCatcher002 and adhiron G5 was confirmed by a size-exclusion chromatography that indicated the formation of a complex after co-incubation of ligand and antigen (Appendix A). ELISA performed using serially diluted antigen concentrations allowed for the calculation of EC50 values of adhirons for the SpyCatcher in the high nM range (Table 2 and Figure 4b). Additionally, we measured the affinity of G5 with alternative methods since different experimental settings could affect the absolute values [40]. In the first case, the *K*D was calculated by employing an electrochemical device. The reproducible data obtained by two independent sets of binding analysis indicated an affinity of roughly 80 nM (Figure 4c and Appendix A). In the second case, the measurement was performed using a dip-in SPR device, and the G5 affinity was estimated to be 87 nM (Appendix A Top). We exploited the same experimental conditions to compare the binding strength of monovalent G5 (G5-APEX2) and bivalent G5 construct (G5-rFc) obtained by the fusion of G5 to a rabbit Fc domain. The data confirmed that the avidity effect allowed increasing the binding apparent affinity from 87 nM to 0.46 nM (Appendix A Bottom). The same G5 adhiron proved to be suitable also for western blotting (Figure 4d), indicating that the adhirons could recognize linear epitopes.

Panning against SARS-CoV-2 Spike RBD resulted in the isolation of 19 clones representing 10 unique adhiron sequences that will be better characterized in a dedicated study (Appendix A). However, such sequences were used together with those of the other isolated adhirons to evaluate some general features of these ligands. 

First, the pI of the selected adhirons was calculated. They varied between 4.73 and 9.95 (Appendix A), and we observed that the ligands selective for each antigen tended to have similar predicted Ip values (slightly acidic for the anti-protoplasts, prevalently basic for the anti-CRP, and neutral for the anti-SpyCatcher and anti-RBD), even though the sequences of the variable loops were highly divergent. Future structural works will help clarify this point and if there is a correlation between adhiron Ip and antigen epitopes, namely, if ligands sharing the Ip interact with the same epitope and Ip outliers with alternative ones. Next, adhirons were sub-cloned into expression vectors in a frame with a 6xHis sequence, suitable for affinity purification, and another tag, chosen according to the downstream application (e.g., mClover, rFc, CysTag, and APEX). About 100 adhiron constructs were expressed in the cytoplasm of BL21(DE3) cells, with clones containing cysteine residues produced in BL21(DE3) but co-expressing sulfhydryl oxidase (SOX) [32] for optimal cytoplasmic oxidizing conditions. Production yields ranged from 2 to 25 mg/L of culture (Figure 5a), with variations being dependent on the tag fused to the adhiron rather than on the adhiron sequence, as clearly shown for the anti-SpyCatcher002 G5 (Figure 5b). Since adhiron G5 contains two cysteine residues, these results showed the robustness of the expression systems based on sulfhydryl oxidase co-expression and also for the production of cysteine-containing adhirons.

The thermal stability of nine adhirons was determined by dynamic scanning fluorometry and revealed that engineered adhirons had, on average, a melting temperature of 59 °C (Figure 5c), which was lower than the parental consensus phytocystatin protein, which is around 100 °C [14]. Therefore, the insertion of the artificial loops substantially decreased the thermal stability of the resulting adhirons, and buffer optimization compensated such a loss only minimally (Appendix A). Furthermore, no evident correlation was observed between Tm and the final yields or the number of cysteines in the adhiron sequence. 

As observed for the production yields, the nature of the tag fused to the adhiron affected its thermal stability, which was also exemplified by the anti-SpyCatcher002 G5 (Figure 5d). Surprisingly, in the case of G5 constructs (Figure 5d), we observed an unexpected inverse correlation between Tm and adhiron production yields that underlined how structural stability was only one component contributing to protein expression. Finally, we assessed the possibility of using thermal purification to separate the most stable adhiron constructs from bacterial contaminant proteins. Specifically, we incubated at increasing temperatures the supernatant fractions of bacteria transformed with B12-SpyTag (Tm 79.84 °C) and G5-CysTag (Tm 92.53 °C), respectively. When incubation temperatures below the Tm values were used, almost pure and monodispersed samples were obtained, as evidenced by SDS-PAGE and gel filtration analyses (Appendix A). Conversely, when we used the temperatures superior to the Tm, it was still possible to recover soluble and relatively pure protein samples, but size-exclusion chromatography indicated potential aggregation and misfolding/degradation of the samples (Appendix A).

## 4. Conclusions

The presented data showed that the synthetic library prepared hypermutating two loops of a phytocystatin scaffold was useful for identifying functional ligands for a set of antigens with widely different characteristics. Currently, the lack of structural data does not allow for evaluating whether the presence of cysteines in the adhiron loops results in wider paratope diversity with respect to affimer sequences with no cysteine, but our data confirmed the versatility of the ligands recovered from this library, allowing for the selections against demanding antigens such as protoplasts as well as soluble proteins with very diverse structures, such as SpyCatcher002, a target that was not possible to recognize with nanobody libraries. The 18 amino acids randomly introduced in the hypervariable loops in place of the six original residues represent as much as 20% of the whole sequence. Indeed, such a relevant modification affected the thermal stability of the scaffold, yet the tested adhirons all displayed a Tm greater than 55 °C and could always be produced in considerable quantities. Importantly, our data showed that the presence of cysteines in the adhiron loops did not introduce stability problems nor impaired their folding in the bacterial cytoplasm. As in the case of plant cyclotides [41], the presence of cysteines might become a means to widen the structural diversity of ligand collections. 

## Figures and Tables

**Figure 1 biomolecules-13-01533-f001:**
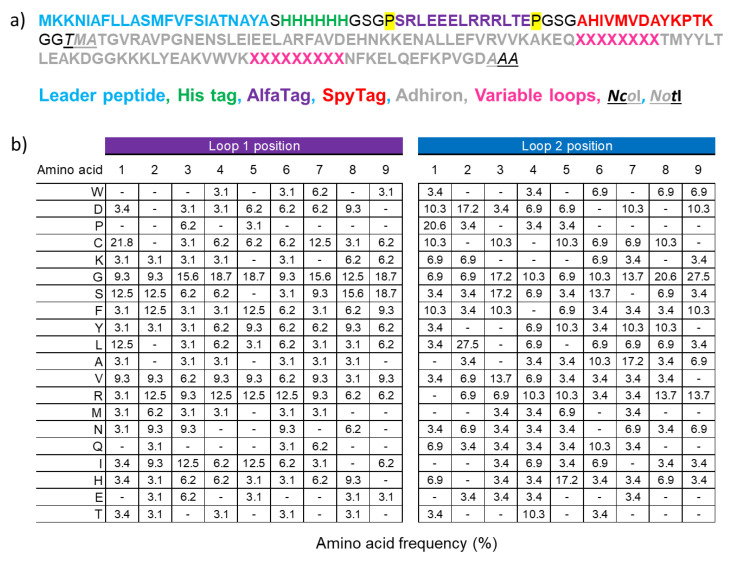
Library characteristics. (**a**) The adhiron sequences have been cloned into a phagemid that has been designed for display on the major phage coat protein pIII. The adhiron-encoding sequence was cloned in a frame with a secretion peptide, a 6xHis tag to aid protein purification, an AlfaTag, and a SpyTag for protein detection as well as rapid and quantitative bioconjugation. The *Nco*I and *Not*I cloning sites were included for rapid subcloning of adhirons into bacterial expression vectors. Proline residues isolating the ALFAtag are highlighted in yellow. (**b**) The actual amino acid frequency of the two hypervariable loops was inferred by the analysis of randomly selected 192 clones.

**Figure 2 biomolecules-13-01533-f002:**
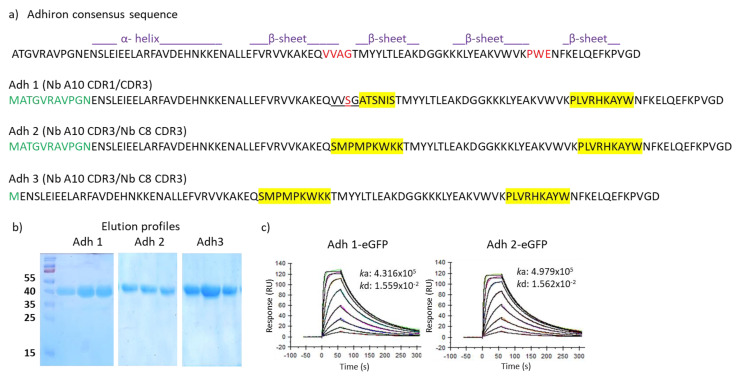
Evaluation of alternative adhiron scaffolds. (**a**) Three different adhiron scaffolds were used to accommodate variable combinations of two CDRs (in yellow) recovered from nanobodies possessing high binding affinity for HER2. Sequences in green indicate the unfolded N-terminal residues of the phytocystatin scaffold, whereas underlined amino acids in the first loop of Adh1 show wild-type sequence of the parental phytocystatin scaffold. Adhirons were expressed as fusions to eGFP, purified by IMAC and gel filtration. Sample purity of the fractions corresponding to the elution peaks was assessed by SDS-PAGE (**b**), and binding constant was determined by SPR (**c**).

**Figure 3 biomolecules-13-01533-f003:**
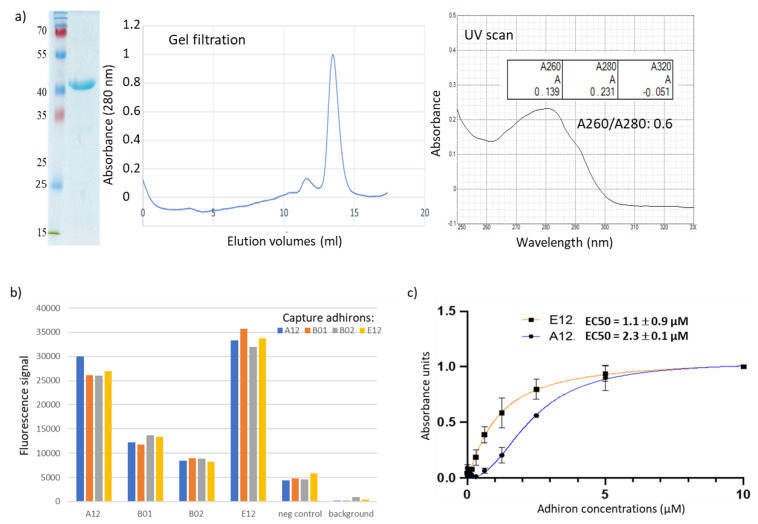
Characterization of anti-pea protoplast adhirons. (**a**) Representative SDS-PAGE, size exclusion chromatography, and UV scan of the protoplast-targeting adhiron E12. The presence of contaminants was ruled out by SDS-PAGE imaging, and adhiron E12 was eluted as a homogeneous and monodisperse peak from gel filtration chromatography. The absence of aggregation was detected by UV scan from 250 to 330 nm (right panel). (**b**) Fluorescent sandwich ELISA. Protoplasts were immobilized on the wells of a 96-well plate to capture adhirons. Following protoplast immobilization, their presence was detected using fluorescently labeled adhirons in solution (A12, B01, B02, and E12), an irrelevant nanobody (neg control), or no binder at all (background). (**c**) The binding affinity of anti-protoplast adhirons was calculated according to the experimental data obtained using concentrations in the range between 10 µM and 5 nM.

**Figure 4 biomolecules-13-01533-f004:**
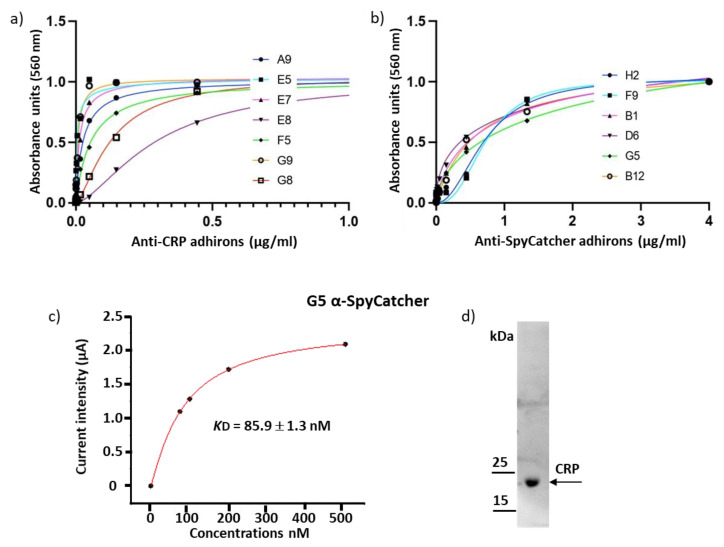
Binding constant of adhirons specific to the soluble antigens CRP and SpyCatcher002. (**a**) Adhirons targeting CRP were fused to APEX and used in an ELISA at concentrations ranging between 4 µM and 23 pM. (**b**) APEX-fused anti-SpyCatcher002 adhirons were used in an ELISA assays at concentrations between 4 µM and 23 pM. (**c**) Affinity of G5 for SpyCatcher002 was measured by chronoamperometry. Experiments were performed in triplicate. (**d**) CRP detection by western blot using the G5 anti-CRP adhiron.

**Figure 5 biomolecules-13-01533-f005:**
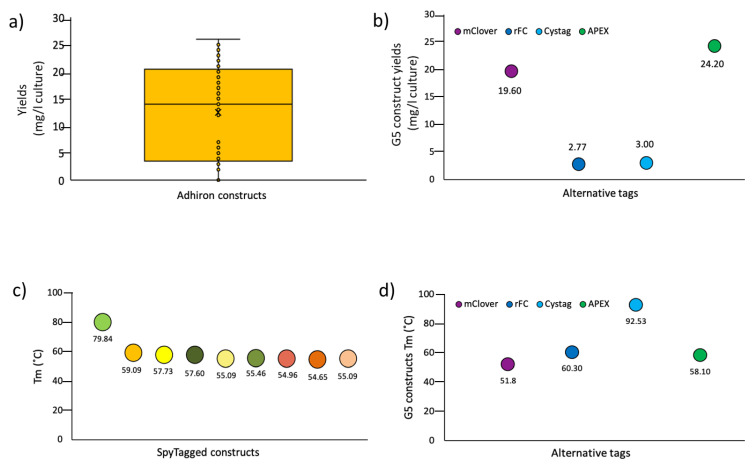
General features of functional adhirons isolated from the synthetic library. (**a**) Yield distribution of different adhiron constructs. The comparison of several clones and constructs obtained from different panning suggests that good yields can be reached, provided that many alternatives are analyzed. (**b**) Effect of the fusion tag on the accumulation of soluble adhiron G5. (**c**) Comparison of the melting temperature of a set of Spy-tagged adhirons (from left to right: B12; E7; H6; H4; C1; F8; E11; A6; G12); (**d**) Effect of the fusion tag on the Tm of different G5 constructs.

**Table 1 biomolecules-13-01533-t001:** Dissociation constants for the adhirons selected against CRP. The EC50 values were determined by ELISA using serial dilutions of each adhiron. The experiments were repeated twice, and data are mean ± SD.

	Anti-CRP						
	A9	E5	E7	E8	F5	G8	G9
EC50 (nM)	25.9 ± 3.0	3.6 ± 0.2	12.9 ± 3.0	295 ± 39.0	55.3 ± 39.0	132 ± 54.0	7.4 ± 0.4

**Table 2 biomolecules-13-01533-t002:** Dissociation constants for the adhirons selected against SpyCatcher002. Binding affinity has been determined by ELISA using serial dilutions of each adhiron. EC50 values have been calculated from data obtained by three independent experiments ± SD.

	Anti-SpyCatcher002					
	B1	B12	D6	F9	G5	H2
EC50 (nM)	577 ± 109	611 ± 70	356 ± 106	727 ± 117	739 ± 263	719 ± 91

## Data Availability

Data are contained within this article or in Appendix A.

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
