# Peer review of "Biological Applications of Synthetic Binders Isolated from a Conceptually New Adhiron Library"

_biomolecules, 2023, doi:10.3390/biom13101533_

Round 1
Reviewer 1 Report
The manuscript titled - "Biological applications of synthetic binders isolated from a 2 conceptually new Adhiron library" has been written well but I have some minor concerns regarding the data presentation:
Line 25- It should be mg/L of culture (Capital L).
Authors might want to explain the relationship between binding affinity and KD.
It would be better for the readers to understand if you can provide a table of all the primers used.
Author Response
Reviewer 1.
The manuscript titled - "Biological applications of synthetic binders isolated from a 2 conceptually new Adhiron library" has been written well but I have some minor concerns regarding the data presentation:
The authors wish to thank the reviewer for her/his suggestions and comments.
Line 25- It should be mg/L of culture (Capital L).
The authors are not aware about the preferred standard of the journal. Since lowercase “l” has been used in the rest of the text, no modification has been introduced now, with the aim of maintaining the consistency throughout the manuscript. However, the authors will modify the text whether the journal will indicate a specific preference.
Authors might want to explain the relationship between binding affinity and KD.
The term “binding affinity” has been used to describe intermolecular interactions in a larger sense with respect to KD (binding constant) that refers only to very specific experimental conditions. In the new version, the term “binding constant” has been used instead of binding affinity when appropriate.
It would be better for the readers to understand if you can provide a table of all the primers used.
As suggested, a Supplementary Table (new Table S1) has been created that reports all the used primers.
Reviewer 2 Report
Here, the manuscript entitled “Biological applications of synthetic binders isolated from a conceptually new Adhiron library” was reviewed. In general, this manuscript developed an Adhiron phage display library that allows the presence of cysteines in the hypervariable loops and successfully panned it against antigens possessing different characteristics. The data in this article is detailed and comprehensive, effectively substantiating the conclusions. Moreover, this is the first report of a panning on plant protoplasts using a library of recombinant ligands, which demonstrates a certain level of innovation. I would like to recommend it to be acceptable for publication after addressing some issues as follows:
1) In the abstract of this paper: firstly, it is recommended to meticulously attend to the employment of conjunctions and transitional lexemes, with the intent of augmenting both the cohesiveness and the seamless logical trajectory of the narrative, particularly within the "results" part; secondly, I suggest the authors endeavor to establish a conspicuous demarcation between the domain of findings and the sphere of discussion. Evidently, there is a tendency for the discourse enclosed within the "discussion" section to more closely resemble the characteristics typically associated with the "results" section. Consequently, a judicious refinement in this regard would substantially enhance the scholarly rigour of the presentation.
2) In the background and within the "Results & Discussion" section of this manuscript, the authors compare and discuss the Adhirons with nanobodies. However, I am interested in elucidating whether Adhirons can be classified as a subclass of cyclotides—an intricate assemblage of plant-derived peptides characterized by their cyclic head-to-tail framework and a central cystine knot motif. It would be of great scholarly significance to explore the parallels and differentials between Adhirons and cyclotides, accentuating both their merits and demerits. Incorporating such a comparative analysis would inherently enhance the comprehensiveness of the article.
3) Kindly ensure meticulous adherence to established writing conventions. For instance, it is imperative to maintain uniformity, as exemplified by the correspondence between "eGFP" as presented in line 302 and "EGFP" as presented in line 311. It is of paramount importance that the complete nomenclature be employed upon the initial mention of "EGFP" within the article, subsequently permitting the utilization of abbreviations for subsequent references.
4) Moreover, I implore your attention to the aesthetic aspects of the visual representations provided in the figures. Notably, in the context of Figure 4, it is incumbent upon the presentation to exhibit a consistent width for the axis lines.
Author Response
Here, the manuscript entitled “Biological applications of synthetic binders isolated from a conceptually new Adhiron library” was reviewed. In general, this manuscript developed an Adhiron phage display library that allows the presence of cysteines in the hypervariable loops and successfully panned it against antigens possessing different characteristics. The data in this article is detailed and comprehensive, effectively substantiating the conclusions. Moreover, this is the first report of a panning on plant protoplasts using a library of recombinant ligands, which demonstrates a certain level of innovation. I would like to recommend it to be acceptable for publication after addressing some issues as follows:
The authors thank the reviewer for having provided not only practical directions but also points for more extensive reflections. We hope (and expect) that the structural investigations we just started will give us some solid information to proceed in the understanding of the modalities necessary to rationally modify cysteine-dependent polypeptide structures.
1) In the abstract of this paper: firstly, it is recommended to meticulously attend to the employment of conjunctions and transitional lexemes, with the intent of augmenting both the cohesiveness and the seamless logical trajectory of the narrative, particularly within the "results" part; secondly, I suggest the authors endeavor to establish a conspicuous demarcation between the domain of findings and the sphere of discussion. Evidently, there is a tendency for the discourse enclosed within the "discussion" section to more closely resemble the characteristics typically associated with the "results" section. Consequently, a judicious refinement in this regard would substantially enhance the scholarly rigour of the presentation.
Thank you for having underlined the deficiencies of the Abstract. The new version should better match your suggestions and appears, at least to the authors, clearer and better articulated.
2) In the background and within the "Results & Discussion" section of this manuscript, the authors compare and discuss the Adhirons with nanobodies. However, I am interested in elucidating whether Adhirons can be classified as a subclass of cyclotides—an intricate assemblage of plant-derived peptides characterized by their cyclic head-to-tail framework and a central cystine knot motif. It would be of great scholarly significance to explore the parallels and differentials between Adhirons and cyclotides, accentuating both their merits and demerits. Incorporating such a comparative analysis would inherently enhance the comprehensiveness of the article.
This is a really interesting point. Indeed, we explored also other short plant proteins/polypeptides to identify useful scaffolds. At the moment, due to the lack of structural data, we cannot really argument on the subject, but we added a sentence in the Conclusion section to link our present knowledge to a future perspective in which the smart design of disulfide bonds might contribute to multiply the “paratope shapes” and, consequently, the possibility to tune protein functions by means of modified 3D scaffolds.
3) Kindly ensure meticulous adherence to established writing conventions. For instance, it is imperative to maintain uniformity, as exemplified by the correspondence between "eGFP" as presented in line 302 and "EGFP" as presented in line 311. It is of paramount importance that the complete nomenclature be employed upon the initial mention of "EGFP" within the article, subsequently permitting the utilization of abbreviations for subsequent references.
The authors apologize for not having respected the consistency rules. The manuscript has been checked carefully and no further similar inaccuracies should be present.
4) Moreover, I implore your attention to the aesthetic aspects of the visual representations provided in the figures. Notably, in the context of Figure 4, it is incumbent upon the presentation to exhibit a consistent width for the axis lines.
Figure 4 has been improved according to the suggestions.
Reviewer 3 Report
The paper “Biological applications of synthetic binders isolated from a conceptually new Adhiron library” by D’Ercole et al. is devoted to construction and characterization of new artificial binding proteins using an Adhiron scaffold. Such binders are gaining growing attention because of their small size, high thermal stability, possibility of production in bacteria and straightforward combination with different tags. The authors produced a synthetic library, isolated efficient binders using panning against various antigens and analyzed their properties. They demonstrated the utility of the described approach which has several advantages in comparison with other popular scaffolds. The purpose of the work is clearly stated and the paper is written in a classic and convincing style. The experiments have been carried out properly in most parts.
Minor points
line 91 The Adhiron sequences were finally cloned in a specifically designed phagemid – Please indicate the name of a phagemid.
line 135 What type of the magnetic beads was used? The related question is about the coating principle (line 140).
line 161-2 saturated with 400 μl of PBS – not clear
Fig 2 – Provide names (a-c) and explanations for different panels. On panel Elution profiles, the numbering and the legends for different lanes should be provided.
Table 1 – The data are also presented in the text (line 387) and on Fig. 3. Therefore, this table which contains only 2 values is not necessary.
line 481, 483 Ip should be pI
Fig 5 – Panel a is not informative. On line 493, yields estimation is provided and the values for several binders are presented elsewhere in the text. Statistical analysis is meaningless in this case because yields for different proteins are compared.
line 507 parental consensus cystatin protein – above, the name of a scaffold was phytocystatin. Please specify.
Author Response
The paper “Biological applications of synthetic binders isolated from a conceptually new Adhiron library” by D’Ercole et al. is devoted to construction and characterization of new artificial binding proteins using an Adhiron scaffold. Such binders are gaining growing attention because of their small size, high thermal stability, possibility of production in bacteria and straightforward combination with different tags. The authors produced a synthetic library, isolated efficient binders using panning against various antigens and analyzed their properties. They demonstrated the utility of the described approach which has several advantages in comparison with other popular scaffolds. The purpose of the work is clearly stated and the paper is written in a classic and convincing style. The experiments have been carried out properly in most parts.
The authors wish to thank the reviewer for the encouraging comments and for the very precise and technical suggestions provided to improve the quality of our contribution.
Minor points
line 91 The Adhiron sequences were finally cloned in a specifically designed phagemid – Please indicate the name of a phagemid.
The information has been provided and the sentence slightly modified to make clearer the content.
line 135 What type of the magnetic beads was used? The related question is about the coating principle (line 140).
The authors apologize for not having mentioned the bead type (M-450 epoxy). The protocol has been optimized over the years (corresponding to roughly 40 panning performed with other phage display libraries). Although the used protein amounts might be different from those suggested by the manufacturer, we experimentally found that the indicated conditions were the most suitable for panning.
line 161-2 saturated with 400 μl of PBS – not clear
The amount of PBS was added to the wells. The sentence has been corrected accordingly.
Fig 2 – Provide names (a-c) and explanations for different panels. On panel Elution profiles, the numbering and the legends for different lanes should be provided.
The Figure 2 has been reorganized accordingly and the legend updated to integrate the missing information.
Table 1 – The data are also presented in the text (line 387) and on Fig. 3. Therefore, this table which contains only 2 values is not necessary.
The Table 1 has been removed, as suggested, and the SD data relative to the EC50 values, previously present in the Table 1, were inserted in the text (line 383).
line 481, 483 Ip should be pI
Thank you for having spotted the mistake, it has been corrected.
Fig 5 – Panel a) is not informative. On line 493, yields estimation is provided and the values for several binders are presented elsewhere in the text. Statistical analysis is meaningless in this case because yields for different proteins are compared.
We can totally understand the reviewer’s point and agree that the representation is “unconventional” but wished to provide, in visual form, an information that is important for practitioners. It is true that it does not make sense, statistically speaking, comparing different proteins and constructs, but it is useful to know what I can expect when I start a project. This piece of information can suggest, for instance, to start with some/several alternative constructs and clones because I know that the yield variability can be pretty wide, but I’ve good chances to find at least a productive clone/construct. Therefore, we’d like to keep this panel and added some information in the legend to clarify what is the aim of mixing oranges and apples.
line 507 parental consensus cystatin protein – above, the name of a scaffold was phytocystatin. Please specify.
The correct definition is phytocystatin, the text has been amended.